# Analysis of Floral Color Differences between Different Ecological Conditions of *Clematis tangutica* (Maxim.) Korsh

**DOI:** 10.3390/molecules28010462

**Published:** 2023-01-03

**Authors:** Xiaozhu Guo, Gui Wang, Juan Li, Jiang Li, Xuemei Sun

**Affiliations:** 1Academy of Agriculture & Forestry, Qinghai University, Xining 810016, China; 2Laboratory for Research and Utilization of Germplasm Resources in Qinghai Tibet Plateau, Xining 810016, China

**Keywords:** *Clematis tangutica* (Maxim.) Korsh, different flower colors, ecological environment, flavonoids

## Abstract

The *Clematis tangutica* (Maxim.) Korsh. is a wild flowering plant that is most widely distributed on the Qinghai–Tibet Plateau, with beautiful, brightly colored flowers and good ornamental properties and adaptability. In diverse natural environments, the blossom color of *C. tangutica* (Maxim.) Korsh. varies greatly, although it is unclear what causes this diversity. It was examined using UPLC-MS/MS and transcriptome sequencing for the investigation of various compounds, differentially expressed genes (DEGs), and flavonoid biosynthesis-related pathways in two flowers in two ecological settings. The results showed that a total of 992 metabolites were detected, of which 425 were differential metabolites, mainly flavonoid metabolites associated with its floral color. The most abundant flavonoids, flavonols and anthocyanin metabolites in the G type were cynaroside, isoquercitrin and peonidin-3-*O*-glucoside, respectively. Flavonoids that differed in multiplicity in G type and N type were rhoifolin, naringin, delphinidin-3-*O*-rutinoside, chrysoeriol and catechin. Rhoifolin and chrysoeriol, produced in flavone and flavonol biosynthesis, two flavonoid compounds of *C. tangutica* (Maxim.) Korsh. with the largest difference in floral composition in two ecological environments. In two ecological environments of flower color components, combined transcriptome and metabolome analyses revealed that *BZ1-1* and *FG3-1* are key genes for delphinidin-3-*O*-rutinoside in anthocyanin biosynthesis, and *HCT-5* and *FG3-3* are key genes for rhoifolin and naringin in flavonoid biosynthesis and flavone and flavonol. Key genes for chlorogenic acid in flavonoid biosynthesis include *HCT-6*, *CHS-1* and *IF7MAT-1*. In summary, differences in flavonoids and their content are the main factors responsible for the differences in the floral color composition of *C. tangutica* (Maxim.) Korsh. in the two ecological environments, and are associated with differential expression of genes related to flavonoid synthesis.

## 1. Introduction

Flower color, a vital floricultural trait in ornamental plants, is the result of the accumulation of flavonoids, carotenoids and betalains [1]. Flavonoids are a kind of secondary metabolites that are widely distributed in plants and are among the most important pigments, the different composition and content of which can lead to different colors in plants [2,3]. Meanwhile, flavonoids are one of the most important pigments in ornamental petals and fruits, such as *Chrysanthemum morifolium* [4], *Dahlia variabilis* [5], *Rosa hybrida* L. [6], *rubus* fruits [7] and *Nymphaea * ”*Blue Bird*” [8]. However, the composition of flavonoids may vary considerably between the differently colored petals of the same species. Study has shown that the detection and analysis of the flower color components of pink and white chrysanthemums reveal that the two anthocyanins in pink chrysanthemums are the main flavonoids, while the flower color components in white chrysanthemums are flavonoids rather than anthocyanins [9]. Analysis of pigments in purple, red, orange, yellow and white *Lycoris longituba* petals showed that only cyanidin-3-xylosylglucoside (Cy3XyGlc) anthocyanins were present in all purple, red and orange petals, while no anthocyanins were detected in the white and yellow petals [10]. Because among flavonoids, anthocyanin belongs to the red series and controls the pink to blue-violet flower color, the other flavonoids belong to the pure yellow series, among which chalcones and aurones are deep yellow, and flavones, flavonols and flavanones are light yellow or nearly colorless [11]. As flavonoids, anthocyanin have highly characteristic C6-C3-C6 carbon skeletons and similar biosynthetic sources [12]. Due to the instability of anthocyanin, most of them exist in glycosylated form [13]. Currently, there are several hundred anthocyanin [14]; the six most common anthocyanin are: pelargonidin(Pg) [15], cyanidin(Cy) [16], delphinidin(Dp) [17], peonidin(Pn) [18], petunidin(Pt) [19] and malvidin(Mv) [20]. Among them, Pg, Cy and Dp are the three main anthocyanin [21]. Studies have shown that Pn is derived from Cy and Pt and Mv from Dp [21].

The pathway of flavonoid biosynthesis in plants is clear [22,23,24]. The starting material of this pathway was phenylalanine, which was catalyzed by phenylalanine ammonia-lyase (PAL), cinnamic acid 4-hydroxylase (C4H) and 4 coumarate CoA ligase (4CL) to produce p-coumaroyl-CoA. Subsequently, p-coumaroyl-CoA was catalyzed by chalcone synthase (CHS), and chalcone isomerase (CHI) to generate chalcones and flavanones in turn, forming the basic structure of anthocyanin. Flavanones were then catalyzed by flavanone 3-hydroxylase(F3H) to produce dihydroflavonols, a branch of the pathway of anthocyanin biosynthesis. Dihydroflavonols can be catalyzed to form dihydroquercetin, dihyromyricetin flavonoids, kaempferol, pelargonidin and flavonols, which are subsequently catalyzed by dihydroflavonol reductase (DFR) to form colored and unstable floral glycosides such as pelargonidin, cyanidin and delphinidin. Finally, the three anthocyanin are catalyzed by anthocyanidin synthase (ANS), glycosyltransferases (UGTs) and methyltransferases (MT) to form stable anthocyanin [25], which are transported to the vesicles for storage [26].

*C. florida* Thunb is one of the world’s most important horticultural plants, with a wide variety of colors and great ornamental value, and is known as the “Queen of Vines” [27]. *C. tangutica* (Maxim.) Korsh. [28] is a widely distributed plant on the Qinghai–Tibet Plateau, growing in grasslands, thickets and rock piles in alpine areas at altitudes of over 3000 m. It has excellent ecological adaptations such as drought tolerance, cold tolerance and light tolerance, and its beautiful, brightly colored flowers and long flowering period can be used as a vertical greening material. It has high ornamental value. At present, most of the research on it is focused on its medicinal value.

Qinghai has a large altitudinal span, with a wide variation in habitat at different altitudes, and the flowers of Ganqing clematis growing there differ greatly in morphology and color (Figure 1). However, differences in floral color components and their key synthesis genes have not been reported for *C. tangutica* (Maxim.) Korsh., and it is not clear how floral color differences relate to the different ecological environments in which they occur. Therefore, the clarification of the floral pigment composition and the key genes for its biosynthesis in it are of great importance in exploring the adaptation of floral color to adversity as well as in breeding and production. In this study, the flowers of wild *C. tangutica* (Maxim.) Korsh. from two ecological environments were used for qualitative and quantitative analysis of flower metabolites using UPLC-MS/MS to investigate the relationship between flower color and anthocyanin metabolites. Identification of different anthocyanin metabolites and analysis of associated differentially expressed genes (DEGs) were based on metabolome and transcriptome. The study’s findings not only offer potential candidates for the metabolic network of anthocyanin biosynthesis in the flowers, but they also offer useful references for further functional validation of the potential candidates and enrichment of the database of anthocyanin components in Ranunculaceae Juss.

## 2. Results

### 2.1. Analysis of Metabolites in the Flowers of C. tangutica (Maxim.) Korsh

The metabolites were studied in order to clarify the differences in flower color between the two ecological conditions of the *C. tangutica* (Maxim.) Korsh. A total of 992 metabolites were identified in its flowers, of which 177 were flavonoid metabolites (Appendix A). The flavonoid metabolites are grouped into nine categories, including 63 flavonols, 38 flavones, 19 anthocyanin, 19 flavanonols, 13 flavonoid carbonoside, 11 isoflavones, seven flavanones, five chalcones and two flavanols. Of the 992 metabolites, there were 425 differential metabolites. Apart from phenolic acids, flavonoids have the highest number of differential metabolites, with 100 species. The results show that the skeletons of most of the flavonoids in it include apigenin, quercetin, kaempferol and lignan. Most of the flavonoids are *O*-glycosides and only a few are *C*-glycosides. Among the metabolites of flavonoids, quercetin, kaempferol, apigenin, lignan, delphinidin, cyanidin, geranophyllin, genistein and paeoniflorin are abundant in it. Among the differential metabolites, three were more than five-fold higher in the G type than in the N type, namely catechin (31.44-flod), pinobanksin (21.40-flod) and peonidin-3-*O*-glucoside (5.36-flod). In addition, the levels of chlorogenic acid and 5-*O*-caffeoylshikimic acid generated in flavonoid biosynthesis were significantly higher in the G type than in the N type.

A total of 19 anthocyanin were identified in *C. tangutica* (Maxim.) Korsh., including delphinidin, cyanidin, peonidin, petunidin and rosinidin. Of these, the levels of peonidin-3-*O*-glucoside, delphinidin-3-*O*-glucoside (myrtillin) and delphinidin-3,5,3′-Tri-*O*-glucoside in the G type were significantly higher than those in the N type (Figure 2). Thus, several compounds, chlorogenic acid, catechin, pinobanksin and peonidin-3-*O*-glucoside, may play an important role in its flower color.

Metabolome correlation analysis of the samples was carried out to assess the biological replication between samples within the group. The Pearson’s correlation coefficient r (PCR) was used as an indicator for the assessment of biological repeat relevance. The closer the absolute value of R is to 1, the stronger the correlation between the two replicate samples. The results showed that there were large differences between sample groups and small differences within groups (Figure 3). 

The overlap display analysis of the total ion flow plots analyzed by mass spectrometric detection of different QC samples showed high overlap of the curves of the total ion flow for metabolite detection, i.e., the retention time and peak intensity were consistent, indicating good signal stability of the mass spectrometry for the same sample at different times (Figure 4A,B; Appendix A). The high stability of the instrument provides an important guarantee of repeatability and reliability of the data, where N stands for the negative ion mode and P for the positive ion mode. 

Overall, the flavonoids in the N type were higher than the flavonoids in the G type flowers, suggesting that these differential metabolites associated with flower color may be the main reason for the differences in flower color between the two types. 

### 2.2. Differential Genes in Clematis tangutica (Maxim.) Korsh. Flowers in Different Ecological Environments

In order to study the gene expression of the pigment constituents of the flowers of *C. tangutica* (Maxim.) Korsh. in different ecological environments, we sequenced the transcriptome of the G type and N type. The six transcriptome samples produced a total of 41.4 Gb clean data, with each sample reaching 6 Gb or more of clean data and 92% or more of Q30 bases. Comparison and annotation of the clean read yielded 123,331 genes, of which 81,893 were newly identified. There were 32,036 DEGs between the two types. 

To analyze the function of DEGs in two types, KEGG enrichment analysis was performed on their DEGs and differential metabolites, demonstrating the extent of the enrichment of pathways with differential genes at the same time. The KEGG enrichment *p*-value results showed that the most DEGs were enriched in metabolic pathways (ko01100), followed by biosynthesis of secondary metabolites (ko01110), and the rest in order of plant hormone signal transduction (ko04075), phenylpropanoid biosynthesis (ko00940) and carotenoid biosynthesis (ko00906) (Figure 5). There are many DEGs associated with flavonoid synthesis between its two different floral colors, further confirming that the flavonoid and flavonol synthetic pathways and the anthocyanin synthetic pathway are the main metabolic pathways responsible for differences in floral color in different ecologies. 

### 2.3. Expression Patterns of Genes Expressed in Relation to the Flavonoid Synthesis Pathway

To further investigate the differential metabolic network of the G type and N type substances, this study targeted 44 genes related to the flavonoid, anthocyanin, flavonoid and flavonol biosynthetic pathways for gene expression pattern analysis (Figure 6). 

In the flavonoid pathway, cinnamoyl-CoA is used as a precursor substance, in turn, in the presence of trans-cinnamate 4-monooxygenase (CYP73A), chalcone synthase (CHS), chalcone isomerase (CHI), naringenin 3-dioxygenase (F3H) and bifunctional dihydroflavonol 4-reductase/flavanone 4-reductase (DFR) to produce pelargonidin, cyanidin and delphinidin into anthocyanin biosynthesis; in the presence of flavonoid 3’,5’-hydroxylase (CYP75A), CHS, CHI and flavonol synthase (FLS), apigenin and kaempferol are produced to enter flavone and flavonol biosynthesis. In the flavone and flavonol biosynthesis, apigenin and kaempferol are the initial precursors in CYP75A, isoflavone 7-*O*-glucoside-6″-*O*-malonyltransferase (IF7MAT), flavonol-3-*O*-glucoside/galactoside glucosyltransferase (FG3) and flavonol-3-*O*-glucoside L-rhamnosyltransferase (FG2) in the presence of a series of enzymes such as kaempferol 3-sophorotrioside, nictoflorin, chrysoeriol and rhoifolin. 

Through a combined metabolome and transcriptome sequencing analysis, we screened 39 genes in the flavonoid biosynthetic pathway in the G type and N type, which had widely divergent expression patterns. The expression of genes in flavonoid biosynthesis, anthocyanin biosynthesis and flavone and flavonol biosynthesis differed between the two ecological environments, with some genes being highly expressed in the G type and lowly expressed in the N type; for example, *CHS-1*, *CHS-5*, *CHI-1*, *CHI-6*, *HCT-2*, etc. in flavonoid biosynthesis, and *IF7MAT-1*, *IF7MAT-2*, *FG3-1*, etc. in flavone and flavonol biosynthesis. Thus, high expression of genes such as CHS, *CHI*, *HCT* and *IF7MAT* upstream of flavonoid biosynthesis and flavone and flavonol biosynthesis may directly affect the synthesis of downstream metabolites. 

### 2.4. Correlation between Flavonoid Content and Flavonoid Synthesis Gene Expression

To further investigate the key genes in the biosynthetic pathway of the floral components of *C. tangutica* (Maxim.) Korsh., we performed a correlation heat map analysis and correlation network analysis on the expression of genes related to the synthesis of anthocyanin and flavonoids in its floral color, leading to the preliminary identification of key genes in the anthocyanin and flavonoid synthetic pathways. The heat map of gene and metabolite correlations showed some variation in the correlation between genes in the anthocyanin and flavonoid synthesis pathways and anthocyanin and flavonoids, with individual genes in flavonoid biosynthesis, anthocyanin biosynthesis and flavone and flavonol biosynthesis having varying degrees of influence on the synthesis of various flavonoids (Figure 7). 

The G type and N type have significantly different anthocyanin compositions and their synthesis-related genes are mainly *CHS-1*, *BZ1-1*, *BZ1-2*, *BZ1-3*, *HCT-2*, *IF7MAT-1*, *FLS* and other genes. In addition, chlorogenic acid generated in flavonoid biosynthesis was the phenolic acid compound with the largest fold difference between G type and N type pigment components and was most strongly correlated with the *CHS-1* gene. Peonidin-3-*O*-glucoside, generated in the anthocyanin biosynthesis pathway, is the most abundant anthocyanin in its floral pigment composition and is most associated with the *BZ1-3* gene. Isoquercitrin and cynaroside, produced in the flavone and flavonol biosynthesis pathway, are the two compounds with the highest relative flavonoid and flavonol content of G type pigment components. Rhoifolin and chrysoeriol are the two flavonoid compounds with the greatest fold difference between the G type and N type floral components, with rhoifolin and chrysoeriol best correlated with the *FG3-3* and *FG3-1* genes, respectively. 

At the same time, we performed a correlation analysis between the flavonoid biosynthesis, anthocyanin biosynthesis and flavone and flavonol biosynthesis for the generation of significantly different delphinidin-3-*O*-rutinoside, rhoifolin chrysoeriol, naringin, chlorogenic acid and other major flavonoid and anthocyanin differential metabolites for correlation analysis. The results showed that delphinidin-3-*O*-rutinoside was positively correlated with the expression of *FG3-1* and *BZ1-1*, suggesting that these genes play a role in the promotion of anthocyanin biosynthesis. Rhoifolin was positively correlated with the expression of *HCT-2*, *HCT-4*, *HCT-5*, *FG3-3* and *IF7MAT-2*, and naringin was positively correlated with the expression of *FG3-3* and *HCT-5*, suggesting that these genes play a positive regulatory role in flavonoid biosynthesis and flavone and flavonol biosynthesis. Chlorogenic acid was positively correlated with the expression of *CHS-1*, *HCT-6* and *IF7MAT-1*, suggesting that these genes play a positive regulatory role in flavonoid biosynthesis (Figure 8). 

## 3. Discussion

The appearance and color of a plant’s flowers are important indicators of its ornamental value, and all ornamental plants have their own ornamental characteristics [29]. Plant flowers show color differences, generally due to the important role played by metabolites such as anthocyanin, flavonoids or carotenoids, such as cyanidin 3,5-*O*-diglucoside, malvidin 3,5-diglucoside and cyanidin 3-*O*-galactoside in purple *Salvia miltiorrhiza* Bge. Flowers [30]; cyanidin 3-*O*-[2-*O*-(xylosyl)-galactoside] in *Nerium oleander* L. [31], etc. Flavonoids are among the most abundant secondary metabolites in the composition of plant anthocyanins and make an important contribution to plants. Other than that, flavonoids have some resistance to UV rays [32]. We sequenced the metabolome and transcriptome of *C. tangutica* (Maxim.) Korsh. flowers from different ecological environments to clarify the substances that differentiate its floral color and to identify key genes in the flavonoid biosynthesis pathway. The relative content of flavonoid compounds in the N type was higher than that in the G type. Chlorogenic acid, pinobanksin, peonidin-3-*O*-glucoside and myrtillin are the main substances in the floral pigment composition of the G type. Of these, rhoifolin, naringin, chrysoeriol and chlorogenic acid are the flavonoids with the greatest relative differences in the floral pigment composition of two types (chlorogenic acid [33] is a phenolic acid, but it binds to anthocyanin and has a stabilizing effect on them). In the same yellow flower, the concentration of flavonoids in lisianthus (*Eustoma grandiflorum*), *Lathyrus chrysanthus* and *Dianthus caryophyllus* petals is high [34]. The main pigment component in American lotus (*Nelumbo lutea*) is flavonols: quercetin 3-*O*-glucuronide [35]. Flavonoids are the main pigment component in paeonia with yellow flowers [36]. 

Correlation analysis of the flavonoids, related gene expressions and intergroup correlation network analysis of the floral components of *C. tangutica* (Maxim.) Korsh. showed that *BZ1-1* and *PG3-1* are key genes for delphinidin-3-*O*-rutinoside in anthocyanin biosynthesis, *HCT-5* and *FG3-3* are rhoifolin and naringin key genes in flavonoid biosynthesis and flavone and flavonol biosynthesis. *HCT-6*, *CHS-1* and *IF7MAT-1* are key genes for chlorogenic acid in flavonoid biosynthesis. In *Acer truncatum* [37], studies on its different leaf colors revealed that the key genes for the biosynthesis of phenyl propane are *CHS*, *PAL*, *C4H* and *4CL*, the key genes for the biosynthesis of flavonoids are *CHI*, *FLS* and *ANR*, and the key genes for the biosynthesis of anthocyanin are *ANS*, *DFR*, *HCT*, *BZ1* and *GT1*. A study of the molecular mechanism of petal discoloration in Nelumbo nucifera [38] revealed that the key genes for anthocyanin biosynthesis in its petals are *CHS*, *F3H*, *ANS* and *UFGT*. Fifteen genes related to flavonoids were identified in Hibiseu manihot L. [39], including *CHS*, *CYP73A*, *CHI*, *F3H*, *DFR*, *HCT*, *ANR* and others. It is suggested that differential genes such as *CHS*, *HCT*, *PG3* and *BZ1* in its petals may be responsible for the differences in flavonoid composition and their relative content, resulting in differences in the flower color of *C. tangutica* (Maxim.) Korsh. under different growth conditions. 

The color of a plant’s flowers not only aids in insect pollination, but also serves as a mechanism for the plant to react to environmental changes. Plants’ bloom colors additionally have significant commercial importance. For example, they can be used as a raw material for the extraction of food additive colors [40], in cosmetics [41], medical treatment [42], as ornamental plants [43], greenery [44] and so on. The ecological conditions under which the two experimental materials used in the study were grown differed considerably in terms of altitude, temperature, humidity, light intensity, soil type, etc. Combined multiomics analyses are often used to investigate phenotypic and biological process regulation mechanisms in biological models. In this study, the relative content of flavonoids and the genes related to flavonoid synthesis in the floral color components of the *C. tangutica* (Maxim.) Korsh. were analyzed by metabolome and transcriptome sequencing techniques under different growth environments to clarify the differences in its floral color components and to lay the foundation for the study of the molecular mechanism of biosynthesis of different floral color substances as well as the influence of different ecological environments on its floral color as well as the selection and breeding of ornamental *C. tangutica* (Maxim.) Korsh. varieties. However, the causes of the environmental conditions responsible for the differences in flower color between the G and N type need further verification.

## 4. Materials and Methods

### 4.1. Plant Material

Flowers of the wildflower species *C. tangutica* (Maxim.) Korsh. from different ecological environments in Qinghai Province were used as test material (Figure 1) and were harvested from three biological replicates in Hainan Tibetan Autonomous Prefecture, Qinghai Province, G109, Gonghe County (36°70′ N, 99°54′ E; altitude 3507 m) and Xining City, Qinghai Province (36°72′ N, 101°7′ E; altitude 2270 m) (denoted by G type and N type for the flowers of the Gonghe and Xining regions, respectively). After being harvested, they were quickly frozen in liquid nitrogen and then stored in a −80 °C cold storage.

### 4.2. Extraction and Analysis of Flavonoid Metabolites

The samples were removed from the −80 °C freezer and placed in a freeze dryer (Scientz-100 F) for vacuum freeze drying. A grinding machine (MM 400, Retsch, Haan, Germany) (30 Hz, 1.5 min) ground them into powder. A 100 mg of powder was weighed and dissolved in 1.2 mL of 70% methanol extract. It was vortexed every 30 min, each session lasting 30 s, for a total of six vortexes. After centrifugation (12,000 rpm, 10 min), the supernatant was aspirated, and the sample was filtered through a microporous membrane (0.22 μm pore size) and stored in the injection vial for UPLC-MS/MS analysis.

### 4.3. RNA Extraction and Sequencing

RNA extraction, cDNA library construction and sequencing were all done by Metware (http://www.metware.cn/ accessed on 12 January 2021). The total RNA was extracted from flower samples using the RNAprep Pure Plant kit (TIANGEN, Beijing, China). The constructed cDNA libraries were sequenced using the Illumina HiSeq platform. After sequencing was completed, clean reads were obtained by fastp_0.19.3 [45] removing adapter related, containing N and Low quality. *C. tangutica* (Maxim.) Korsh. is a reference-free genome, and clean reads were spliced using Trinity [46] to obtain reference sequences for subsequent analysis.

### 4.4. Transcriptomic Data Analysis

Unigene sequences were compared to the KEGG, NR, Swiss-Prot, GO, KOG, Trembl databases using DIAMOND_v0.9.24.125 [47] BLAST software, and to the Pfam database using HNNER software to obtain annotation information for Unigene. Using bowtie_2 2.3.4.1 [48] of the RSEM_v1.3.1 [49] software, we obtained the transcriptome by splicing Trinity_v2.11.0 as the reference sequence, and the clean reads of the samples were mapped towards the reference sequence. To allow the number of fragments to truly reflect the level of transcript expression, the number of mapped reads and the length of transcripts at the sample site were normalized and FPKM (fragments per kilobase of transcript per million fragments mapped) was used as a measure of the level of transcript or gene expression. 

Differential expression analysis between sample groups was performed by DESeq2_1.22.2 [50,51] using unstandardized gene reads count data to obtain differentially expressed gene sets between the two biological conditions. The reads count of genes was implemented using feature counts. The Benmini–Hochberg method was then used to apply multiple hypothesis correction to the probability of hypothesis testing (*p*-value) to obtain the false discovery rate (FDR). The differential genes were screened for |log2Fold Change| ≥ 1, and FDR < 0.05. The obtained DEGs were annotated to the KEGG (Kyoto Encyclopedia of Genes and Genomes), GO (Gene Ontology) and KOG (Cluster of Eukaryotic Orthologous Groups) databases and enrichment analysis was performed by applying hypergeometric tests. 

### 4.5. Correlation Analysis and Correlation Network Analysis

In order to identify the genes that play a key role in the flavonoid biosynthesis pathway in the different flower colors of glycine clematis flowers, correlation analysis and correlation network analysis were used to establish the flavonoid content and expression levels of genes related to flavonoid synthesis in glycine clematis flowers from both regions. Correlation analysis was performed by R (base package) 3.5.1 (default parameters), where the correlation coefficient > 0.80 and *p* value < 0.05. Correlation network analysis was performed using Spearman of Cytoscape_v3.6.1 to calculate correlations, where the correlation coefficient > 0.95 and the *p*-value < 0.01.

### 4.6. Statistical Analysis

The experimental data were analyzed using SPSS_18.0 with normal distribution validation followed by independent sample t-tests to determine the significance of differences (*p* < 0.05). Hierarchical clustering analysis of metabolites between each sample was carried out using R 3.5.1. After normalization of the data, the heat map was plotted using heatmap (R)_1.0.12.

## Figures and Tables

**Figure 1 molecules-28-00462-f001:**
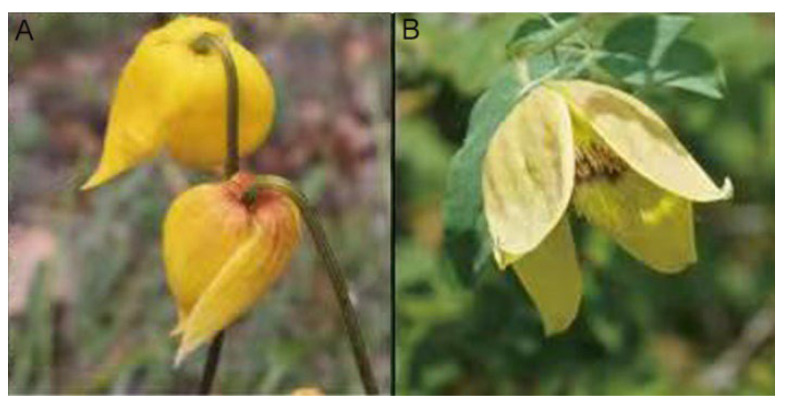
The flower of *Clematis tangutica* (Maxim.) Korsh. (**A**) G type (**B**) N type.

**Figure 2 molecules-28-00462-f002:**
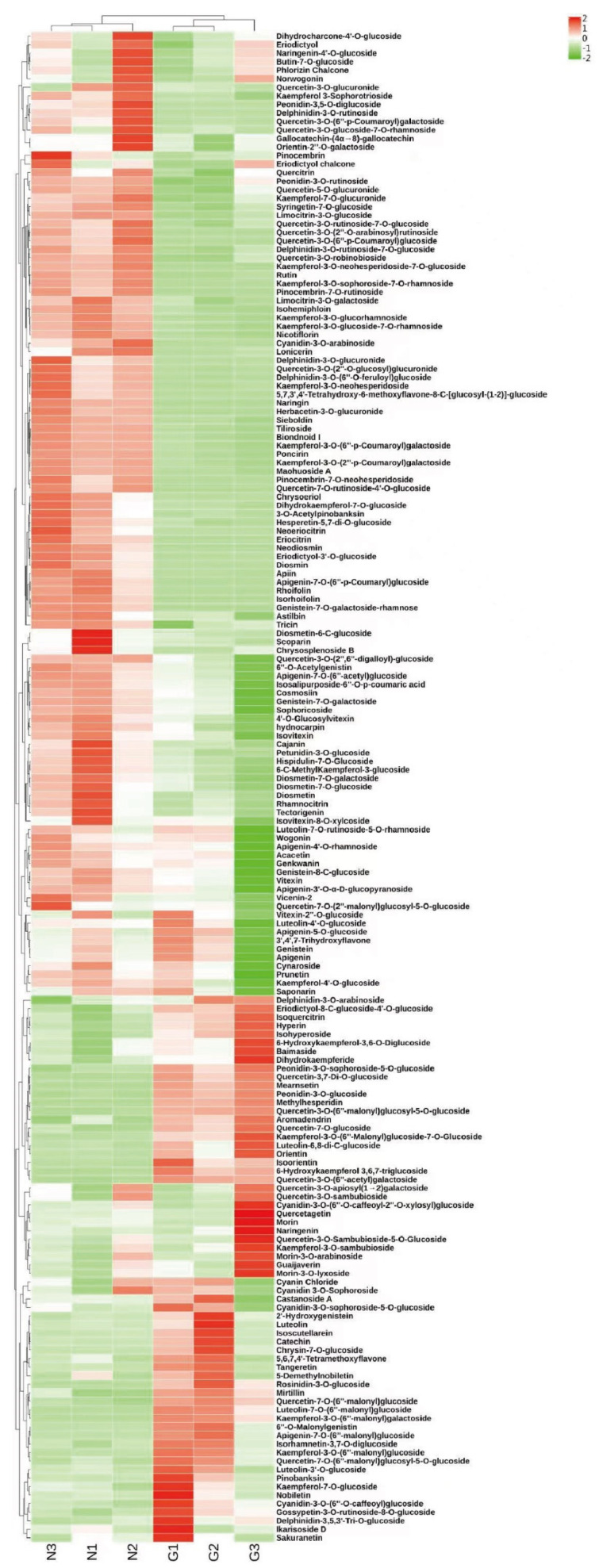
Heat map of the relative content of flavonoids in G type and N type (different colors represent Z-score normalized values for the relative content of metabolites; red represents high compound content, green represents low compound content).

**Figure 3 molecules-28-00462-f003:**
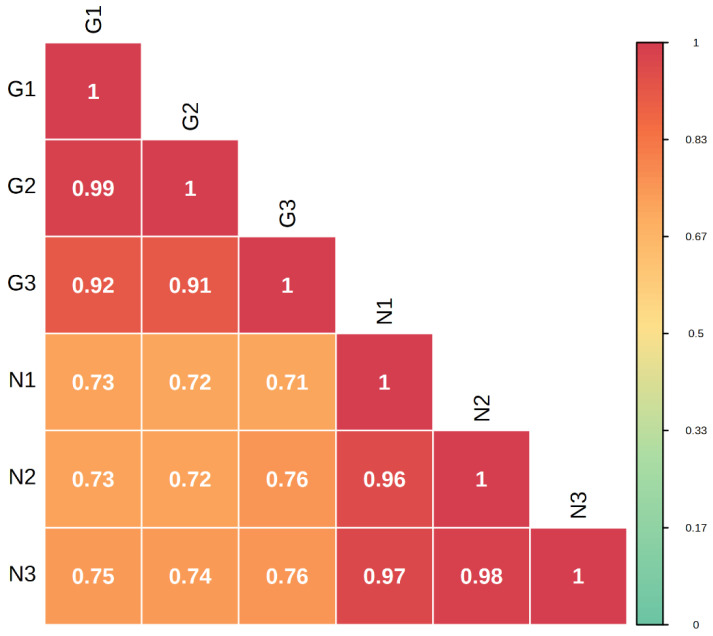
Correlation plot between G type and N type samples.

**Figure 4 molecules-28-00462-f004:**
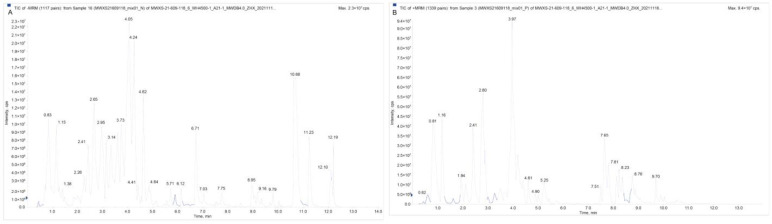
Total ion flow diagram for mixed sample mass spectrometry ((**A**) QC-MS-TIC-N (**B**) QC-MS-TIC-P).

**Figure 5 molecules-28-00462-f005:**
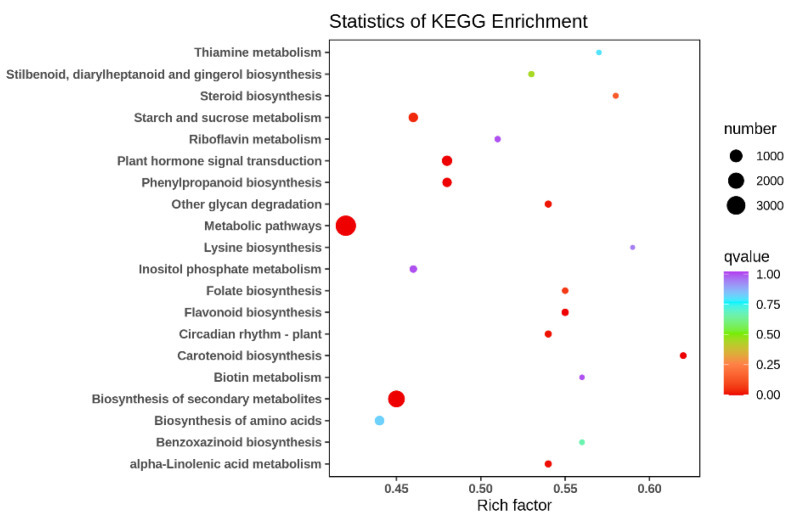
Scatter plot of differential gene KEGG enrichment in G type and N type.

**Figure 6 molecules-28-00462-f006:**
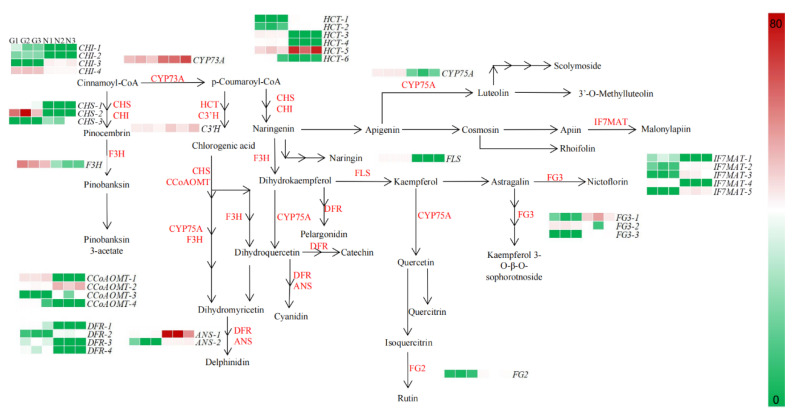
Expression patterns of flavonoid biosynthesis-related genes in flowers of G type and N type.

**Figure 7 molecules-28-00462-f007:**
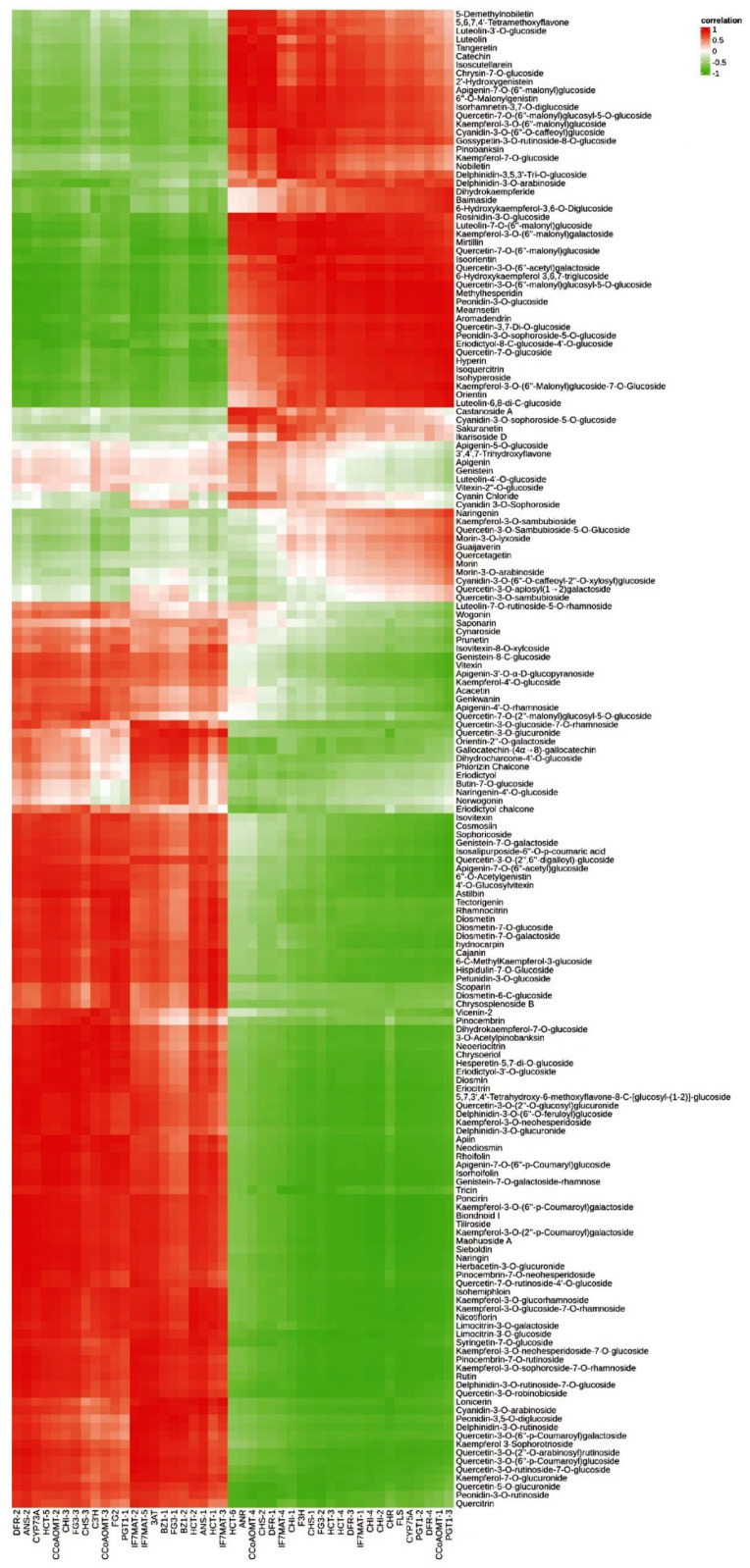
Correlation analysis of flavonoids and flavonoid synthesis genes in flowers of *Clematis tangutica* (Maxim.) Korsh.

**Figure 8 molecules-28-00462-f008:**
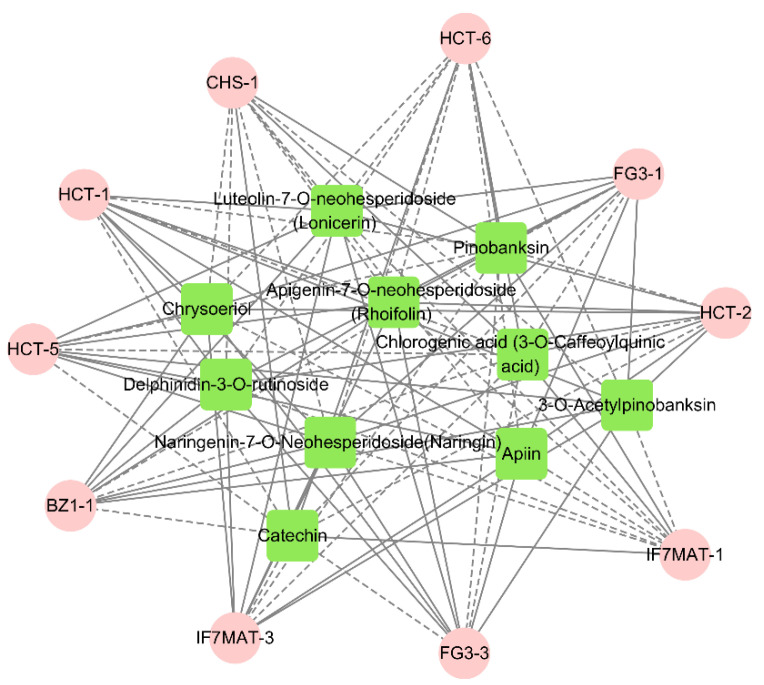
Metabolites and genes combined analysis network (pink represents genes; green represents metabolites. Solid lines represent positive correlations between metabolites and genes; dashed lines represent negative correlations between metabolites and genes. Shorter lines represent stronger correlations between metabolites and genes).

## Data Availability

The datasets are publicly available at NCBI with Sequence Read Archive (SRA) accession: PRJNA893815.

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
