# Peer review of "Analysis of Floral Color Differences between Different Ecological Conditions of *Clematis tangutica* (Maxim.) Korsh"

_molecules, 2023, doi:10.3390/molecules28010462_

Round 1

Reviewer 1 Report

In the present manuscript, the floral color differences of Clematis tangutica (Maxim.) Korsh under different ecological conditions was investigated. This study was interesting and was organized well. Hwoever, there were still some minor problems which should be revised before its publication. Some comments or suggestions were as following:

1, In the title and the text, there were some non-standard format problem. The plant they studied should be expressed as Clematis tangutica (Maxim.) Korsh.

2, When the genus name Clematis occured in the same paper more than once, it shoud be abbrevated as C..

3, Sinece this study was about the floral color differences, why did they do the work about the terpenoid biosynthesis-related pathways? They were Irrelevant.

4, In the term of the anthocyanins, the letter O should be in italic, as well as in quite a lot of flavonoids names.

5, There were too many figures in this manuscript. Figure 2 and Figure 6 were unnecessary. And also. Figure 5 could show more imformation if it could be expressed as a table.

6, In Figure 1, the name of the plant was wrong.

7, There were quite a lot of non-standard format problems in the text. For example, there should be a space between the number the unit.

8, The authors should pay more attention to the so called different ecological conditions”, which were very important for this study, but with quite limited introduction and explanation.

Author Response

The comments and suggestions 1、2、3,  4、6、7 have been modified.

Issue 5: the data in Figure 5 is too cumbersome and therefore I would like to present it as supplementary material

Issue 8: I have added to the discussion section

Thank you very much for your valuable comments

Reviewer 2 Report

The piece of work is interesting and generated some useful information on‘Analysis of floral color differences between different ecological conditions of Clematis tangutica (Maxim.)Korsh. Overall, the experiments were done with sound methods and manuscript was written well to explain the objectives. However, I have some minor issues that are to be addressed before the article being accepted. These are as follows:

1. What  do G1-G3 and N1-N3 indicate in Figure 3 and Figure 4?

2. Line 3-5‘。。。lites such as anthocyanin, flavonoids or carotenoids,such as cyanidin 3,5-O-diglucoside, malvidin 3,5-diglucoside and cyanidin 3-O-galactoside in purple Salvia miltiorrhiza Bge. 。。in the part of Discussion is difficult to follow

3. Material,4.5 Grammar error is found in the last sentence’ where Correlation coefficient > 0.95 and p-value < 0.01.’

4. Only the organ flower can be shown Figure 1,so it should be Flower of lematis tangutica(Maxim.)Korsh.

5..The references No page was found in the Ref.2 and Ref.20 

Author Response

Issue1 

The number of 1、2、3 represents three repeats of G and N type.

Issue 2

It is intended that the important metabolites related to flower colour in plants of different colours are listed in order to elicit the different metabolites that cause different flower colours in Clematis tangutica(Maxim.)Korsh. under different ecological conditions

Issue 3 have been modified.

Issue 4 have been modified.

Issue 5

The Ref.2 is in the 5 line of the Introduction,the Ref.20 is in the 23 line of the Introduction.

Thank you very much for your valuable comments and I would appreciate your criticism if there are any inappropriate points!